# From Noisy Oracles to Useful Constraints: LLM-Guided Constraint Selection for Synthetic Tabular Data

**Tejumade Afonja** [1]   **Joscha Cüppers** [1]   **Mario Fritz** [1]

## Abstract

Structured foundation models increasingly depend on large corpora of high-quality tabular data for pretraining and evaluation. Yet standard deep generative models routinely violate domain constraints, producing structurally implausible samples that are unsuitable for high-stakes downstream tasks. Constraint-aware generation addresses this but requires constraints to be specified upfront, creating a costly annotation bottleneck. We propose `EVS` (*Extract*, *Verify*, *Select*), which removes this bottleneck by treating large language models as scalable but noisy proxies for domain knowledge. `EVS` extracts candidate constraints from column metadata, verifies them against the training data, and selects a utility-preserving subset via binary search in $O(\log K)$ model evaluations without any ground-truth constraint labels. Across three datasets and four LLMs, `EVS` reduces constraint violation rates to near zero while matching the downstream utility of unconstrained generation, and approaches the performance of expert-annotated baselines when the LLM recovers high-quality constraints.

## 1. Introduction

Structured foundation models for tabular data–including models such as ConTextTab (Spinaci et al., 2026), and Tab-DPT (Ma et al., 2024)–are trained on large and diverse collections of tabular datasets. As these models scale, the quality of their training corpora becomes a first-order concern: just as language models benefit from curated web text, tabular foundation models benefit from structurally coherent data. Synthetic tabular data generation has emerged as a promising route for expanding data sharing (Xu et al., 2019; Zhao et al., 2023), data augmentation (Antoniou et al., 2017), and the construction of diverse benchmarking suites (Erick-

son et al., 2026). Yet a fundamental gap persists: deep generative models (DGMs) treat generation as an unconstrained optimisation problem and routinely produce samples that violate *domain* constraints–the predicates over individual rows that the true data-generating process respects by construction. (Stoian et al., 2024) reports violation rates above 95% on several real-world datasets. Samples that violate constraints have zero probability under the correct data-generating process; feeding them to a structured foundation model distorts its learned priors in ways that can be hard to detect and expensive to correct.

**The constraint bottleneck.** Existing constraint aware generative models (Stoian et al., 2024; Zhao et al., 2023) assume that the relevant constraints are provided upfront by a domain expert. This assumption is rarely satisfied in practice: domain constraints remain implicit in expert knowledge, scattered across data dictionaries, or only partially documented. The annotation bottleneck limits constraint-aware synthesis to the small fraction of datasets with unusual complete documentation–precisely the kind of dataset that is already well-represented in foundation model training corpora.

**LLMs as constraint oracles.** Large language models (LLMs) are an increasingly central component of structured data pipelines. They serve as automated annotators (Gilardi et al., 2023), feature engineers (Hollmann et al., 2023), and schema reasoners (Nagarajan & Altman, 2026). Given column names and natural language descriptions, a capable LLM can propose many plausible domain constraints without human annotation. However, raw LLM proposals are noisy: they may include hallucinated boundaries, redundant constraints, or constraints that are jointly harmful to data quality. In our experiments, naive enforcement of all LLM-proposed constraints reduces downstream AUC up to 26 points relative to unconstrained generation.

We propose `EVS` (*Extract*, *Verify*, *Select*), a three-stage pipeline that uses LLMs as noisy oracles whose outputs are filtered and selected using data itself as the arbiter of constraint quality (Figure 1). Our main contributions are:

1. **`EVS` pipeline**: a fully automated method combining

[1]CISPA Helmholtz Center for Information Security, Saarbrücken, Germany. Correspondence to: Tejumade Afonja <tejumade.afonja@cispa.de>.

*Foundation Models for Structured Data ICML Workshop*, 2026.

LLM constraint extraction, data-driven verification, and binary-search selection without ground-truth labels.

2. $O(\log K)$ **selection**: a binary search algorithm that identifies a utility-preserving constraint subset using a one-sided significance test against a multi-run baseline.

3. **Empirical validation**: across three datasets and four LLMs, EVS reduced violation to near zero while preserving downstream utility, outperforming raw and filtered baselines.

## 2. Related Work

**Synthetic tabular data generation.** GAN-based methods (Xu et al., 2019; Zhao et al., 2021), variational autoencoders (Xu et al., 2019), and diffusion models (Kotelnikov et al., 2023) have pushed distributional fidelity close to real data on standard benchmarks. More recently, relational structure between features has been incorporated to improve statistical realism (Liu et al., 2023). However, standard evaluation protocols–marginal distributions, correlation matrices, and downstream classifier utility–are blind to structural plausibility. A model can achieve state-of-the-art on every standard metric while violating the majority of domain constraints, because none of these metrics penalise violations directly. This evaluation gap motivates our work: statistical fidelity is necessary but not sufficient for synthetic data to serve as a valid substitute in high-stakes settings or as training data for structured foundation models.

**Constraint-aware generation.** Hard enforcement methods guarantee that generated samples lie in the feasible region by projecting onto it during training or inference (Stoian et al., 2024). Soft enforcement methods instead penalise violations in the generator loss (Wu et al., 2020; Zhao et al., 2023). Both families share a critical assumption: the constraint set is provided upfront by a human expert. We treat enforcement as a solved subproblem–adopting a similar projection approach of (Stoian et al., 2024)–and focus on the constraint *discovery and selection* problem that prior work did not cover.

**LLMs for structured data tasks.** LLMs have been applied as hypothesis generators, scientific reasoners, and data annotators (Gilardi et al., 2023), and more recently as sources of structured domain knowledge for automated feature engineering (Hollmann et al., 2023) and schema understanding (Nagarajan & Altman, 2026). The consistent finding is that LLM outputs are useful as a starting point but require downstream validation: LLMs hallucinate plausible-sounding but incorrect facts and miss dataset-specific structure that is invisible from metadata alone. EVS instantiates this pattern in the constraint domain, coupling LLM proposals with empirical data-driven filtering and selection.

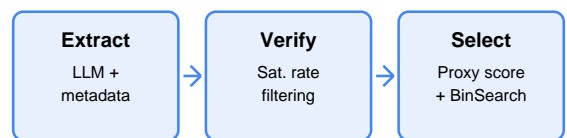

*Figure 1.* The EVS pipeline: Extract candidate constraints from an LLM using column metadata; Verify them against training data; Select a utility-preserving subset via binary search.

## 3. EVS Framework

EVS operates on a dataset with $d_{\text{num}}$ numerical columns. Figure 1 illustrates the three stages.

### 3.1. Extract: LLM Constraint Proposal

Given column names, data types, and natural language descriptions, an LLM is prompted to propose linear constraints of the form $\mathbf{w}^\top \mathbf{y} \geq b$, where $\mathbf{w} \in \mathbb{Z}^{d_{\text{num}}}$ and $b \in \mathbb{R}$. Each constraint must involve at least two variables and be accompanied by a justification and a self-assessed validity label; constraints flagged *Invalid* by the LLM itself are discarded. The prompt explicitly forbids trivially-satisfied constraints (e.g., non-negativity of a single feature) and rescaled duplicates. The resulting candidate set is $\hat{\mathcal{C}} = \{(\mathbf{w}_i, b_i)\}$. We provide the full prompt in Appendix D.7.

### 3.2. Verify: Data-Driven Pruning

Raw LLM proposals often include constraints the training data itself violates, and that contradict one another. The verify stage prunes $\hat{\mathcal{C}}$ to a clean set $\hat{\mathcal{C}}_v$ using two steps.

**Satisfaction-rate filtering.** For each candidate $(\mathbf{w}_i, b_i)$, we compute the empirical satisfaction rate $\rho_i = \frac{1}{n} \sum_{j=1}^{n} \mathbf{1}[\mathbf{w}_i^\top \mathbf{x}_j + b_i \geq 0]$ on the training data. Candidates with $\rho_i < \tau$ (default $\tau = 0.90$) are discarded: they are either hallucinated or too tight relative to the true generating process.

**Redundancy pruning.** Direction-equivalent constraints (weight vectors pointing in the same direction) are consolidated by retaining only the tightest one, since the rest are implied.

### 3.3. Select: Binary Search over Ranked Candidates

Even after verification, jointly enforcing all surviving constraints can degrade synthetic data quality. We therefore need to find the largest subset that does not harm utility.

**Proxy scoring.** Each surviving constraint $i$ is scored by the sum of signed distances from training points to its bound-

ary:

$$s_{ij} = \frac{\mathbf{w}_i^\top \mathbf{x}_j + b_i}{\|\mathbf{w}_i\|_2}, \quad \phi_i = \sum_j |s_{ij}|.$$

Constraints are sorted by ascending $\phi_i$: a lower score means the boundary lies closer to (and is more informative about) the training distribution. This yields a ranked candidate list $\hat{\mathcal{C}}_{\text{pre}}$ of size $K$.

**Binary search.** We assume monotonicity: if the $k$-th constraint is harmful, all lower-ranked constraints are also harmful. The selection task is therefore to find the largest prefix $k^*$ of $\hat{\mathcal{C}}_{\text{pre}}$ such that enforcing that prefix does not significantly reduce downstream AUC. We train the unconstrained model $n_{\text{base}}$ times to obtain a baseline AUC distribution $\mathcal{A}_0$, then binary-search over $k \in \{1, \ldots, K\}$, comparing each $\mathcal{A}_k$ to $\mathcal{A}_0$ via a one-sided Welch's $t$-test at $\alpha = 0.05$. A prefix is accepted if it does not cause a statistically significant AUC drop. This requires $O(\log K)$ full model evaluations.

**Constraint enforcement.** Finally, we describe how we enforce selected constraints $S^*$. We follow the approach proposed by Stoian et al. (2024). Their approach requires constraints to be provided in a reduced backtrack free representation and requires a user-specified variable ordering. We hence introduce an adaption of their approach, CLAMP. CLAMP does not require a specific representation and ordering, allowing us to enforce any extracted satisfiable constraint sets.

To enforce selected constraints $S^*$ we project generated samples onto the feasible region $\mathcal{F} = \{\mathbf{x} : \mathbf{w}_i^\top \mathbf{x} + b_i \geq 0, \ \forall i \in S^*\}$. For each generated sample $\mathbf{x}$, we solve:

$$\mathbf{x}^* = \arg\min_{\mathbf{z} \in \mathcal{F}} \|\mathbf{z} - \mathbf{x}\|_2^2, \tag{1}$$

using CVXPY (Diamond & Boyd, 2016). Following Stoian et al. (2024), gradients are passed through the projection layer during training via the straight-through estimator, allowing the generator to learn representations that take advantage of the enforced constraints rather than treating enforcement as a post-hoc correction.

## 4. Experiments

**Setup.** We evaluate on three datasets: FAULTS[1] (surface defect detection, dense constraints), HELOC[2] (credit scoring, sparse rules), and WIDS[3] (clinical sepsis prediction, 31 expert constraints with full ground truth). Constraint extractors: GPT-5.5 (OpenAI, 2026), GPT-4.1 (OpenAI, 2025), Claude Opus 4 (Anthropic, 2026), and Llama-3.1-8B (Dubey et al., 2024). Generator: CTGAN (Xu et al.,

---

[1]https://www.kaggle.com/datasets/uciml/faulty-steel-plates
[2]https://huggingface.co/datasets/mstz/heloc
[3]https://www.kaggle.com/competitions/widsdatathon2021

---

*Table 1.* Raw and filtered enforcement (AUC / violation rate %).

| Method | FAULTS | | HELOC | | WIDS | |
|---|---|---|---|---|---|---|
| | AUC | Viol. | AUC | Viol. | AUC | Viol. |
| Uncons. | .708 | 84.0 | .714 | 47.5 | .768 | 100.0 |
| *Raw LLM enforcement* | | | | | | |
| GPT-5.5 | .521 | 0.4 | .648 | 1.0 | .718 | 0.0 |
| Claude-4 | .706 | 9.3 | .646 | 0.2 | .717 | 0.0 |
| GPT-4.1 | .528 | 85.7 | .547 | 4.4 | .730 | 0.0 |
| Llama-8B | .713 | 78.5 | .711 | 58.5 | .754 | 100.0 |
| *Filtered (Verify only)* | | | | | | |
| GPT-5.5 | .696 | 0.0 | .664 | 1.2 | .718 | 0.0 |
| Claude-4 | .706 | 9.3 | .621 | 0.3 | .717 | 0.0 |
| GPT-4.1 | .514 | 83.8 | .658 | 2.1 | .730 | 0.0 |
| Llama-8B | .713 | 78.5 | .709 | 53.8 | .754 | 100.0 |

*Table 2.* Full EVS results. *no constraints selected (unconstrained fallback). Expert: human-annotated constraints.

| Method | FAULTS | | HELOC | | WIDS | |
|---|---|---|---|---|---|---|
| | AUC | Viol. | AUC | Viol. | AUC | Viol. |
| Uncons. | .708 | 84.0 | .714 | 47.5 | .768 | 100.0 |
| Expert | .716 | 0.0 | .672 | 0.0 | .728 | 0.0 |
| GPT-5.5 | .713 | 0.0 | .664 | 1.2 | .718 | 0.0 |
| Claude-4 | .706 | 9.3 | .714* | 47.5 | .717 | 0.0 |
| GPT-4.1 | .708* | 84.0 | .658 | 2.1 | .730 | 0.0 |
| Llama-8B | .713 | 78.5 | .709 | 53.8 | .754 | 100.0 |

2019). Metrics: AUC ($\uparrow$) and violation rate ($\downarrow$), averaged over 5 seeds $\times$ 5 synthetic datasets. Details about metrics and datasets in Appendix B.

**Raw and filtered proposals are insufficient (Table 1).** Enforcing all LLM-extracted constraints without selection impacts downstream utility: GPT-5.5 drops from 0.708 (unconstrained) to 0.521 on HELOC. Filtering alone (the Verify stage only) partially recovers utility (0.696), but still underperforms compared with the unconstrained AUC, though it considerably reduces the violation rate from 84% (unconstrained) to 0.4% (raw).

**EVS recovers utility while eliminating violations (Table 2).** The full pipeline simultaneously drives violation rates toward zero and matches or exceeds unconstrained AUC across datasets and LLMs. Crucially, the violation rate has been greatly reduced. On FAULTS, GPT-5.5 EVS achieves 0.713 AUC (vs. 0.708 unconstrained) with 0.0% violations, up from 0.521 under raw enforcement. On WIDS, GPT-5.5, Claude Opus 4, and GPT-4.1 all reach 0.0% violations while maintaining high AUC. Where binary search rejects all candidates (marked *), EVS falls back to unconstrained generation, preserving the utility ceiling. Llama-3.1-8B yields no usable constraints on any dataset, confirming that a capable instruction-tuned extractor is necessary.

**LLM constraint quality (Table 3).** Frontier models recover meaningful structural knowledge: GPT-5.5 and Claude Opus 4 achieve 100% recall on WIDS, and GPT-5.5 reaches

*Table 3.* LLM constraint quality after Verify (vs. expert ground truth). Rec. = recall, Prec. = precision, Acc. = accuracy, Filt. = constraints removed by Verify, Top = all true positives appear at head of proxy-score ranking.

| Dataset | LLM | Rec. | Prec. | Acc. | Filt. | Top |
|---------|-----|------|-------|------|-------|-----|
| FAULTS | GPT-5.5 | 1.00 | 0.36 | 0.36 | 5 | ✓ |
| | Claude-4 | 0.75 | 0.60 | 0.50 | 0 | ✓ |
| | GPT-4.1 | 0.00 | 0.00 | 0.00 | 2 | ✗ |
| | Llama-8B | 0.00 | 0.00 | 0.00 | 0 | ✗ |
| HELOC | GPT-5.5 | 0.71 | 0.45 | 0.38 | 1 | ✗ |
| | Claude-4 | 0.86 | 0.67 | 0.60 | 1 | ✓ |
| | GPT-4.1 | 0.86 | 0.42 | 0.40 | 2 | ✗ |
| | Llama-8B | 0.00 | 0.00 | 0.00 | 1 | ✗ |
| WIDS | GPT-5.5 | 1.00 | 0.62 | 0.62 | 0 | ✓ |
| | Claude-4 | 1.00 | 0.70 | 0.70 | 0 | ✓ |
| | GPT-4.1 | 1.00 | 1.00 | 1.00 | 0 | ✓ |
| | Llama-8B | 0.00 | 0.00 | 0.00 | 0 | ✗ |

100% recall on FAULTS. GPT-4.1 achieves perfect precision on WIDS, which may reflect memorisation of its published ground truth. We verify against a simulated TRIATHLON dataset (see Appendix D.8); GPT-5.5 retains 64% recall, confirming genuine semantic reasoning. Llama-3.1-8B scores 0 on all metrics, underscoring the importance of extractor quality. Crucially, the proxy score places true-positive constraints at the top of the ranked list for GPT-5.5 and Claude Opus 4 on most datasets (✓ in Top column), supporting the monotonicity assumption that underpins the binary search.

## 5. Discussion and Conclusion

**Data as the arbiter of constraint quality.** The central lesson of EVS is that data should be the arbiter of whether a proposed constraint is useful. A constraint is useful if enforcing it reduces violations without reducing utility, and uninformative for the task otherwise. This framing explains both why raw LLM proposals underperform (they are not validated against any data signal) and why the Verify-only baseline is insufficient (satisfaction rate alone does not distinguish tight, informative constraints from loose, redundant ones). The proxy score and binary search together provide a computationally efficient mechanism for making this data-driven judgement at scale.

**Implications for structured foundation model pretraining.** Tabular foundation models are pretrained on heterogeneous corpora of real-world datasets. When a dataset is unavailable due to privacy constraints, synthetic surrogates are used in its place. Our results show that unconstrained generators produce surrogates in which 50–100% of rows violate known domain constraints, which is far from the true data-generating distribution. Replacing these with EVS-generated data reduces violations to near zero while preserving distributional fidelity, as measured by downstream AUC, marginal distributions, and feature correlations (see Appendix D.9). Pretraining on structurally valid synthetic data avoids the risk of distorting the model's learned priors with factually impossible rows. Beyond pretraining, EVS-generated data can serve as constraint-compliant augmentation for fine-tuning and evaluation benchmarks, broadening the applicability of structured foundation models to domains where real data is scarce or sensitive.

**Limitations and future directions.** EVS currently handles linear inequality constraints over continuous features. Functional dependencies over categorical features (e.g., zip code determines state) and multi-row or temporal constraints require fundamentally different enforcement mechanisms and are left to future work. The binary search relies on a monotonicity assumption–that if the $k$-th constraint is harmful, all lower-ranked constraints are also harmful–which may not hold when constraints interact in complex ways. The iterative selection phase requires repeated model training, which is computationally expensive for large datasets or long-training generators; future work could amortise this cost using cheaper proxy evaluations. Finally, EVS's success is bounded by the quality of the LLM extractor: for domains with highly specialised or implicit constraints, even frontier models may recover insufficient signal.

**Conclusion.** We presented EVS, a three-stage pipeline that uses LLMs as scalable but noisy proxies for domain knowledge to automate constraint discovery for synthetic tabular data generation. EVS extracts candidate constraints from column metadata, verifies them using empirical satisfaction rates and redundancy checks, and selects a utility-preserving subset through binary search, without ground-truth labels or manual annotation. Across three datasets and four LLMs, EVS substantially reduces violation rates while preserving downstream utility, outperforming raw and filtered enforcement baselines and approaching expert-annotation quality when the LLM recovers ground-truth constraints. These results suggest that LLMs can serve as scalable but noisy proxies for domain expertise, provided their outputs are coupled with empirical validation and careful subset selection.

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

# A. Appendix

# B. Experimental Setup

## B.1. Datasets

We evaluate on three tabular datasets spanning finance, manufacturing, healthcare, cybersecurity, and sports simulation. All datasets and their associated constraints are sourced from the benchmark repository of Stoian et al. (2024); we refer the reader there for full preprocessing and constraint derivation details. Table 4 summarises their statistics and constraint characteristics.

*Table 4.* Dataset statistics and constraint characteristics. **Num** = numerical features, **Cat** = categorical features, **Total** = total features, **Cst** = constraints, **Min / MaxV** = min / max variables in a single constraint, **CV** = constrained variables. FAULTS is a multi-class classification task; all others are binary classification. Triathlon is included exclusively to assess whether LLMs can propose valid constraints for a dataset potentially absent from their training data.

| Dataset | Split Size | | | Features | | | Constraints | | | |
|---------|-------|------|------|-----|-----|-------|-----|-----------|----|-------------|
| | Train | Test | Val | Num | Cat | Total | Cst | Min / MaxV | CV | Cst with CV¿2 |
| HELOC | 7897 | 220 | 1755 | 23 | 1 | 24 | 7 | 2 / 2 | 10 | 0 |
| FAULTS | 1553 | 195 | 193 | 27 | 1 | 28 | 4 | 2 / 2 | 7 | 0 |
| WIDS | 22256 | 6956 | 5564 | 108 | 1 | 109 | 31 | 2 / 2 | 62 | 0 |
| Triathlon | 1600 | 200 | 200 | 12 | 2 | 14 | 13 | 2 / 4 | 11 | 2 |

# C. Metrics

# D. Evaluation Metrics

## D.1. Per-sample Constraint Satisfaction Rate (Cons. Sat.)

measures how well generated or observed samples adhere to a predefined set of constraints. For each datapoint, we compute the fraction of applicable constraints whose violation magnitude falls below a small tolerance threshold (e.g $\delta = 1e - 2$), ignoring constraints that are not defined for that sample. The Constraint Satisfaction is then obtained by averaging this fraction across all data points and reporting it as a percentage.

Let $s_{ij} \in \{0, 1\}$ indicate whether sample $x_i$ satisfies constraint $c_j$. The constraint satisfaction rate (Cons. Sat.) is defined as:

$$\text{Cons. Sat.} = \frac{1}{N} \sum_{i=1}^{N} \left( \frac{1}{M_i} \sum_{j \in \mathcal{A}_i} s_{ij} \right) \times 100$$

where $N$ is the number of samples and $M$ is the number of constraints and $\mathcal{A}_i$ is the set of applicable (non-missing) constraints per sample $i$. Constraint Satisfaction measures the average fraction of constraints satisfied per sample.

## D.2. Per-samples Constraint Violation Rate (Viol. Rate)

measures the proportion of samples that fail to satisfy atleast one constraint, providing a strict sample-level assessment of feasibility. Specifically, a sample is considered valid only if it satisfies all constraints simultaneously, otherwise, it is counted as a violation. The metric is computed as the complement of the joint constraint satisfaction rate and is reported as a percentage.

Let $z_i \in \{0, 1\}$ indicate whether sample $x_i$ violates at least one constraint:

$$z_i = \begin{cases} 1 & \text{if } \exists j \in \{1, \ldots, M\} \text{ such that } s_{ij} = 0, \\ 0 & \text{otherwise.} \end{cases} \tag{2}$$

The percentage of samples violating constraints is defined as:

$$\text{Viol. Rate} = \frac{1}{N} \sum_{i=1}^{N} z_i \times 100.$$

### D.3. Marginal Distribution (Marginal)

measures how well the synthetic data matches the real data at the level of each individual feature, ignoring the dependencies between variables. For categorical features $j \in \mathcal{C}_c$, we compute the Total Variation Distance (TVD) between the empirical categorical distributions of the real and synthetic data, which quantifies the absolute difference between probability masses across all categories.

Total Variation Distance (TVD):

$$\text{TVD}_j = \frac{1}{2} \sum_{k \in \mathcal{X}_j} \left| P_j(k) - Q_j(k) \right|,$$

where $P_j$ and $Q_j$ are the empirical distributions of real and synthetic data.

For continuous features $j \in \mathcal{C}_n$, we compute the Kolmogorov-Smirnov (KS) statistic, which measures the maximum discrepancy between the empirical cumulative distribution functions of the two datasets.

$$\text{KS}_j = \sup_x \left| F_j^{(r)}(x) - F_j^{(s)}(x) \right|,$$

The overall marginal distribution score is defined as the average discrepancy across all features:

$$\text{Marginal} = \frac{1}{d} \left( \sum_{j \in \mathcal{C}_c} \text{TVD}_j + \sum_{j \in \mathcal{C}_n} \text{KS}_j \right).$$

### D.4. Correlation Difference (Corr Diff)

evaluate the preservation of feature dependencies by measuring the discrepancy between pairwise correlation structures of the real and synthetic data.

Let $C^{(r)}$ and $C^{(s)}$ denote the empirical correlation matrices of the real and synthetic datasets, respectively, computed using Pearson (or an alternative) correlation. We define the absolute correlation difference matrix as:

$$D = \left| C^{(r)} - C^{(s)} \right|, \quad D_{ii} = 0,$$

where diagonal entries are excluded.

We report the mean and maximum correlation differences as:

$$\text{MeanCorrDiff} = \frac{1}{d(d-1)} \sum_{i \neq j} D_{ij},$$

$$\text{MaxCorrDiff} = \max_{i \neq j} D_{ij},$$

where $d$ is the number of features.

### D.5. Membership Inference Attack (MIA)

evaluates privacy leakage using a classifier-based membership inference attack where an attacker is trained to distinguish between members (training data) and non-members (holdout data). The trained attacker is then applied to synthetic samples to estimate how similar they are to training data.

For synthetic data $X_{\text{synth}} = \{x_i\}_{i=1}^N$, we report the mean membership probability:

$$\text{synth\_membership\_prob\_mean} = \frac{1}{N} \sum_{i=1}^N p_\theta(x_i),$$

which quantifies the average likelihood that synthetic samples are classified as training members.

To capture high-risk samples, we define a threshold $\tau$ as the $q$-th percentile of membership probabilities on the holdout set:

$$\tau = \text{Percentile}_q \left( \{p_\theta(x) : x \in X_{\text{holdout}}\} \right).$$

The number and fraction of high-risk synthetic samples are then defined as:

$$\text{high\_risk\_samples} = \sum_{i=1}^N \mathbf{1}\left[p_\theta(x_i) > \tau\right] \tag{3}$$

$$\text{high\_risk\_ratio} = \frac{1}{N} \times \text{high\_risk\_samples} \tag{4}$$

### D.6. Downstream Utility

We evaluate the utility of synthetic data by measuring how well models trained on synthetic data generalize to real data. Let $D_s = (X_s, y_s)$ denote the synthetic dataset, and $D_v = (X_v, y_v)$ and $D_t = (X_t, y_t)$ denote real validation and test sets, respectively.

A predictive model $f_\theta$ is trained on the synthetic data:

$$f_\theta = \text{Train}(X_s, y_s),$$

and evaluated on real data using standard performance metrics:

$$\text{Utility} = \text{Score}(f_\theta(X_t), y_t),$$

where $\text{Score}(\cdot)$ depends on the task.

For classification tasks, we use metrics such as Accuracy, F1-score, and ROC-AUC:

$$\text{AUC} = \text{ROC-AUC}(y_t, f_\theta(X_t)).$$

To ensure robustness, we evaluate multiple model families and hyperparameter configurations. We consider Decision Tree Classifier, Logistic Regression, Random Forest Classifier, and XGBClassifier for binary and multiclass classification.

We select the best-performing model on validation data:

$$f^* = \arg\max_{f_\theta \in \mathcal{M}} \text{Score}(f_\theta(X_v), y_v),$$

and report its performance on the test set as the final utility score.

### D.7. Prompts

We prompt the LLM with the dataset name, column names, data types, and any available natural-language column descriptions, and instruct it to return linear inequality constraints over continuous features in a structured format that can be parsed directly into raw constraint set. The full prompt and an example output from GPT-5.5 is detailed below.

**Constraint Generation Prompt**

```
You are an expert in linear inequality constraints for tabular data.
RULES:
- Output ONLY constraints of the form w^T y >= b
- Each constraint MUST be written in the form:
  <linear expression> >= b
- Use ONLY operator ">="
- Variables:  y_0 ...  y_n
- NO variables allowed on the right-hand side
- Integer coefficients only
- At least TWO variables per constraint
- Do NOT negate or rescale previous constraints
- Do NOT output JSON or Markdown
- If unsure, output nothing
FORMATTING RULES:
- No spaces between coefficients and variables (e.g., "2y_5")
Dataset:  {dataset_name}
Column mapping for the numerical columns only:
{mapping}
{column_description}
CRITICAL CONSTRAINT RULES (MUST FOLLOW):
- DO NOT output constraints that are always true due to non-negativity
  (e.g., y_i >= b, y_i + y_j >= b)
- ONLY include hard constraints
- ALWAYS prefer shorter constraints
- DO NOT use any specific knowledge of the dataset -- only the semantics of the
 feature
- Constraints must be domain relevant

 JUSTIFICATION RULES:
- Each constraint MUST be followed immediately by its own justification
- Justify ONLY the constraint directly above
- Output if constraint is Valid or Invalid based on justification
Output format:
START_OF_OUTPUT
<linear constraint>
Justification:  <justification>
Validity:  Valid | Invalid
 ...
```

**Example LLM Output (GPT-5.5 - FAULTS Dataset)**

```
START_OF_OUTPUT
-y_0+y_1>=0
Justification:  The maximum X coordinate cannot be smaller than the minimum X
coordinate for the same fault region.
Validity:  Valid

-y_2+y_3>=0
Justification:  The maximum Y coordinate cannot be smaller than the minimum Y
coordinate for the same fault region.
Validity:  Valid

y_10-y_1>=0
Justification:  An X coordinate on the plate cannot extend beyond the conveyor
length that bounds the plate along that direction.
Validity:  Valid

y_0-y_1+y_5>=0
Justification:  The X-direction perimeter span of a fault must cover at least the
difference between its maximum and minimum X coordinates.
Validity:  Valid

y_2-y_3+y_6>=0
Justification:  The Y-direction perimeter span of a fault must cover at least the
difference between its maximum and minimum Y coordinates.
Validity:  Valid

-y_8+y_9>=0
Justification:  The maximum luminosity within a fault region cannot be smaller than
the minimum luminosity within the same region.
Validity:  Valid

y_7-y_9>=0
Justification:  The sum of luminosities over all pixels in a fault region must be at
least the largest single-pixel luminosity in that region.
Validity:  Valid

-y_11-y_12>=-1
Justification:  A steel plate cannot simultaneously be of two mutually exclusive
steel types A300 and A400.
Validity:  Valid

-y_15-y_16>=-2
Justification:  Empty index and square index are normalized measures, so their
combined value is bounded by their individual normalized upper limits.
Validity:  Valid

-y_17-y_20>=-2
Justification:  Outside X index and outside global index are ratio-based normalized
measures, so their combined value is bounded by their individual normalized upper
limits.
Validity:  Valid

-y_14-y_26>=-2
Justification:  Edges index is a ratio-based index and sigmoid of areas is a
sigmoid-normalized value, so their combined value is bounded by their individual
normalized upper limits.
Validity:  Valid
```

*Table 5.* LLM constraint quality after Verify for Simulated Data (vs. expert ground truth). Rec. = recall, Prec. = precision, Acc. = accuracy, Filt. = constraints removed by Verify, Top = all true positives appear at head of proxy-score ranking.

| Dataset | LLM | Rec. | Prec. | Acc. | Filt. | Top |
|---|---|---|---|---|---|---|
| TRIATHLON | gpt-5.5 | 0.64 | 0.88 | 0.58 | 1 | × |
| | llama-8b | 0.00 | 0.00 | 0.00 | 8 | × |

*Table 6.* Results on the FAULTS dataset. We report utility, privacy, and membership-inference attack metrics, including violation rate, constraint satisfaction, F1, weighted F1, AUC, marginal distribution distance, correlation difference, MIA probability, and high-risk membership rate. ∗gpt-4.1 falls back to unconstrained results since none of the constraints were selected.

| Method | Viol. Rate ↓ | Cons. Sat. ↑ | Utility | | | Marginal ↓ | Corr Diff ↓ | Privacy Mia Attack | |
|---|---|---|---|---|---|---|---|---|---|
| | | | F1 ↑ | wF1 ↑ | AUC ↑ | | | MIA Prob ↓ | High Risk ↓ |
| Real data | 0.0 ± 0.0 | 75.0 ± 0.0 | 0.808 ± 0.000 | 0.810 ± 0.000 | 0.936 ± 0.000 | 0.049 ± 0.000 | 0.038 ± 0.000 | 0.897 ± 0.000 | 0.379 ± 0.000 |
| Unconstrained | 84.0 ± 8.1 | 62.0 ± 4.8 | 0.262 ± 0.017 | 0.263 ± 0.018 | 0.708 ± 0.011 | 0.231 ± 0.029 | 0.196 ± 0.010 | 0.867 ± 0.004 | 0.002 ± 0.000 |
| Expert | 0.0 ± 0.0 | 79.7 ± 2.5 | 0.293 ± 0.028 | 0.295 ± 0.027 | 0.716 ± 0.016 | 0.240 ± 0.020 | 0.204 ± 0.005 | 0.839 ± 0.018 | 0.002 ± 0.000 |
| claude-opus-4-7 | 9.3 ± 9.6 | 77.3 ± 2.4 | 0.284 ± 0.028 | 0.284 ± 0.029 | 0.708 ± 0.012 | 0.230 ± 0.012 | 0.195 ± 0.015 | 0.871 ± 0.002 | 0.002 ± 0.001 |
| gpt-4.1∗ | 84.0 ± 8.1 | 62.0 ± 4.8 | 0.262 ± 0.017 | 0.263 ± 0.018 | 0.708 ± 0.011 | 0.231 ± 0.029 | 0.196 ± 0.010 | 0.867 ± 0.004 | 0.002 ± 0.000 |
| gpt-5.5 | **0.0 ± 0.0** | 79.3 ± 1.0 | 0.292 ± 0.021 | 0.292 ± 0.021 | 0.713 ± 0.008 | 0.233 ± 0.018 | 0.218 ± 0.012 | 0.841 ± 0.010 | 0.002 ± 0.000 |
| llama-8b | 78.5 ± 7.3 | 51.2 ± 4.1 | 0.297 ± 0.029 | 0.297 ± 0.028 | 0.709 ± 0.019 | 0.212 ± 0.016 | 0.204 ± 0.012 | 0.870 ± 0.003 | 0.003 ± 0.001 |

## D.8. Simulated Dataset

## D.9. Full results

Tables 6 - 8 report full `EVS` selection results across all datasets and LLMs.

*Table 7.* Results on the HELOC dataset. We report utility, privacy, and membership-inference attack metrics, including violation rate, constraint satisfaction, F1, weighted F1, AUC, marginal distribution distance, correlation difference, MIA probability, and high-risk membership rate. ∗gpt-4.1 falls back to unconstrained results since none of the constraints were selected.

| Method | Viol. Rate ↓ | Cons. Sat. ↑ | Utility | | | Marginal ↓ | Corr Diff ↓ | Privacy Mia Attack | |
| | | | F1 ↑ | wF1 ↑ | AUC ↑ | | | MIA Prob ↓ | High Risk ↓ |
| --- | --- | --- | --- | --- | --- | --- | --- | --- | --- |
| Real data | $9.0 \pm 0.0$ | $98.0 \pm 0.0$ | $0.744 \pm 0.000$ | $0.707 \pm 0.000$ | $0.775 \pm 0.000$ | $0.047 \pm 0.000$ | $0.044 \pm 0.000$ | $0.864 \pm 0.000$ | $0.772 \pm 0.000$ |
| Unconstrained | $47.5 \pm 4.2$ | $91.9 \pm 0.9$ | $0.709 \pm 0.030$ | $0.666 \pm 0.021$ | $0.714 \pm 0.021$ | $0.077 \pm 0.008$ | $0.122 \pm 0.009$ | $0.812 \pm 0.001$ | $0.125 \pm 0.015$ |
| Expert | $0.0 \pm 0.0$ | $100.0 \pm 0.0$ | $0.679 \pm 0.047$ | $0.631 \pm 0.041$ | $0.672 \pm 0.039$ | $0.081 \pm 0.008$ | $0.115 \pm 0.004$ | $0.809 \pm 0.001$ | $0.089 \pm 0.012$ |
| claude-opus-4-7 | $0.3 \pm 0.1$ | $100.0 \pm 0.0$ | $0.639 \pm 0.026$ | $0.579 \pm 0.026$ | $0.621 \pm 0.036$ | $0.087 \pm 0.009$ | $0.124 \pm 0.007$ | $0.807 \pm 0.002$ | $0.076 \pm 0.015$ |
| gpt-4.1∗ | $47.5 \pm 4.2$ | $91.9 \pm 0.9$ | $0.709 \pm 0.030$ | $0.666 \pm 0.021$ | $0.714 \pm 0.021$ | $0.077 \pm 0.008$ | $0.122 \pm 0.009$ | $0.812 \pm 0.001$ | $0.125 \pm 0.015$ |
| gpt-5.5 | $1.2 \pm 0.4$ | $99.8 \pm 0.1$ | $0.657 \pm 0.059$ | $0.621 \pm 0.061$ | $0.664 \pm 0.078$ | $0.088 \pm 0.009$ | $0.121 \pm 0.007$ | $0.808 \pm 0.001$ | $0.079 \pm 0.017$ |
| llama-8b | $47.9 \pm 7.1$ | $92.0 \pm 1.7$ | $0.709 \pm 0.009$ | $0.646 \pm 0.018$ | $0.695 \pm 0.022$ | $0.077 \pm 0.011$ | $0.123 \pm 0.008$ | $0.810 \pm 0.001$ | $0.106 \pm 0.014$ |

*Table 8.* Results on the WIDS dataset. We report utility, privacy, and membership-inference attack metrics, including violation rate, constraint satisfaction, F1, weighted F1, AUC, marginal distribution distance, correlation difference, MIA probability, and high-risk membership rate.

| Method | Viol. Rate ↓ | Cons. Sat. ↑ | Utility | | | Marginal ↓ | Corr Diff ↓ | Privacy Mia Attack | |
| | | | F1 ↑ | wF1 ↑ | AUC ↑ | | | MIA Prob ↓ | High Risk ↓ |
| --- | --- | --- | --- | --- | --- | --- | --- | --- | --- |
| Real data | $1.0 \pm 0.0$ | $100.0 \pm 0.0$ | $0.367 \pm 0.000$ | $0.419 \pm 0.000$ | $0.830 \pm 0.000$ | $0.019 \pm 0.000$ | $0.021 \pm 0.000$ | $0.826 \pm 0.000$ | $0.849 \pm 0.000$ |
| Unconstrained | $100.0 \pm 0.0$ | $73.5 \pm 1.5$ | $0.305 \pm 0.027$ | $0.352 \pm 0.027$ | $0.747 \pm 0.029$ | $0.298 \pm 0.008$ | $0.101 \pm 0.008$ | $0.720 \pm 0.001$ | $0.351 \pm 0.024$ |
| Expert | $0.0 \pm 0.0$ | $100.0 \pm 0.0$ | $0.275 \pm 0.025$ | $0.320 \pm 0.025$ | $0.728 \pm 0.035$ | $0.280 \pm 0.004$ | $0.084 \pm 0.006$ | $0.725 \pm 0.003$ | $0.413 \pm 0.041$ |
| gpt-5.5 | $0.0 \pm 0.0$ | $100.0 \pm 0.0$ | $0.268 \pm 0.016$ | $0.313 \pm 0.015$ | $0.718 \pm 0.022$ | $0.275 \pm 0.008$ | $0.090 \pm 0.012$ | $0.726 \pm 0.004$ | $0.421 \pm 0.061$ |
| claude-opus-4-7 | $0.0 \pm 0.0$ | $100.0 \pm 0.0$ | $0.261 \pm 0.010$ | $0.306 \pm 0.009$ | $0.717 \pm 0.022$ | $0.276 \pm 0.010$ | $0.097 \pm 0.008$ | $0.726 \pm 0.003$ | $0.435 \pm 0.034$ |
| gpt-4.1 | $0.0 \pm 0.0$ | $100.0 \pm 0.0$ | $0.276 \pm 0.025$ | $0.320 \pm 0.025$ | $0.730 \pm 0.035$ | $0.280 \pm 0.004$ | $0.084 \pm 0.006$ | $0.725 \pm 0.003$ | $0.413 \pm 0.041$ |
| llama-8b | $100.0 \pm 0.0$ | $70.7 \pm 2.0$ | $0.303 \pm 0.038$ | $0.349 \pm 0.038$ | $0.754 \pm 0.026$ | $0.288 \pm 0.012$ | $0.091 \pm 0.007$ | $0.721 \pm 0.002$ | $0.358 \pm 0.027$ |

