# OpenReview forum: "From Noisy Oracles to Useful Constraints: LLM-Guided Constraint Selection for Synthetic Tabular Data"
_ICML.cc/2026/Workshop/FMSD — FMSD @ ICML 2026 Poster_

### Official Review · Reviewer_HgAm · 2026-05-20
**Clean idea, narrow validation - workshop-acceptable but needs broader experiments and an honest look at the utility trade-off.**

**Rating:** 6
**Confidence:** 3

**Review:**

One-line summary: EVS uses LLMs to propose domain constraints for synthetic tabular data, then verifies them against training data and selects a utility-preserving subset via O(log K) binary search - eliminating the expert-annotation bottleneck in constraint-aware generation.

Strengths:
1.Tackles a real and well-motivated bottleneck (expert constraints are rarely available) with a clean three-stage pipeline; the TRIATHLON contamination check and Llama-3.1-8B negative result are nice touches.
2.Strong empirical evidence that raw LLM enforcement hurts (up to 26 AUC drop) while full EVS recovers utility and drives violations near zero, approaching expert-annotation quality on capable LLMs.

Weaknesses:

1. Narrow validation: only CTGAN as generator, three datasets, and the monotonicity assumption underlying binary search is never empirically tested;
2. Underplayed trade-off: expert and EVS constraints actually reduce AUC vs. unconstrained on HELOC and WIDS.

---

### Official Review · Reviewer_Stza · 2026-05-21
**A sound proof-of-concept for automated constraint discovery, marred by an overstated utility claim and an underpowered statistical test.**

**Rating:** 6
**Confidence:** 4

**Review:**

**Summary:**

The paper addresses the bottleneck of expert annotation in constraint-aware synthetic tabular data generation. It proposes EVS (Extract, Verify, Select), a pipeline where an LLM proposes constraints, which are then data-verified and selected via a binary search to preserve downstream utility. Selected constraints are enforced during generation using an adapted projection layer.

**Strengths:**

- Tackles a genuine, well-motivated bottleneck and explicitly scopes its contribution as a delta over prior enforcement mechanisms.
- The "data as arbiter" design is sound, successfully demonstrating that raw, unfiltered LLM enforcement is actively harmful and requires data-driven verification.
- High transparency: includes the full constraint-generation prompt, a complete worked LLM output, and honestly reports the failure of smaller models (Llama-3.1-8B).

**Weaknesses:**

- The central utility claim in the abstract ("matches the downstream utility of unconstrained generation") is contradicted by the paper's own tables, which show utility drops on 2 out of 3 datasets.
- The Select stage relies on an underpowered statistical foundation (failing to reject a null hypothesis using a Welch's t-test over 5 seeds is treated as equivalence).
- The method lacks a no-LLM statistical-mining baseline to prove the LLM oracle actually adds value over basic data bounds.
- Uncited near-neighbors (arXiv:2503.02161, arXiv:2508.02601) and an unvalidated monotonicity assumption for the binary search.

---

### Official Review · Reviewer_mfpG · 2026-05-22
**A step towards automated constraint discovery for synthetic data**

**Rating:** 6
**Confidence:** 2

**Review:**

Strengths
- The paper effectively proposes an alternative to manual annotation through constraint-aware generation, moving toward scalable data generation and curation.
- The verify stage rightly treats LLMs as flawed reasoning engines, using empirical satisfaction rates to prune hallucinations and redundancies.

Areas for Improvement
- The claim of matching unconstrained utility is contradicted by the WIDS dataset, where full EVS introduces a 5-point drop in downstream AUC (0.718 vs. 0.768). However, this drop is mirrored by the human Expert baseline (0.728). The paper could be strengthened by an analysis (statistical/geometrical) on why unconstrained performed better here.
- The pipeline is currently limited to linear inequalities over numerical features. Scaling this to complex applied contexts requires handling functional dependencies, categorical rules, and temporal constraints.
- The workshop explicitly requests cost/latency reporting. The binary search requires O(log K) full model evaluations (CTGAN training) per dataset. This iterative retraining overhead is acknowledged but not quantified. Adding total compute time or cost comparisons against expert annotation is necessary.
- Generation is limited to CTGAN. Testing EVS against modern structured foundation models (e.g., TabPFN, TabDDPM) would better align with the workshop's core focus.
- Standard downstream AUC is used to measure utility. Including domain-specific metrics or measuring the impact of enforced constraints on minority class performance would strengthen the empirical validation.

Justification of Score

The paper presents a solid, relevant approach to improving synthetic data quality for structured modeling. The methodology is sound, and the transition from noisy LLM outputs to verified constraints is well-executed. Addressing the missing compute metrics and testing beyond basic linear continuous constraints will significantly elevate its impact.